# Is retina affected in Huntington's disease? Is optical coherence tomography a good biomarker?

**Pavel Dusek**[1], **Ales Kopal**[1,2], **Michaela Brichova**[3], **Jan Roth**[1], **Olga Ulmanova**[1], **Jiri Klempir**[1], **Jana Lizrova Preiningerova**[1] *

**1** Department of Neurology and Center of Clinical Neuroscience, First Faculty of Medicine Charles University and General University Hospital in Prague, Prague, Czech Republic, **2** Department of Neurology, Regional Hospital Pardubice, Pardubice, Czech Republic, **3** Department of Ophthalmology, First Faculty of Medicine Charles University and General University Hospital in Prague, Prague, Czech Republic

☯ These authors contributed equally to this work.

* jana.lizrova@vfn.cz

**Data Availability Statement:** The data that support the findings of this study are available in the GitHub repository (https://doi.org/10.5061/dryad. ncjsxksxr). Analysis source code may be found in

## Abstract

### Aim of the study

Comparative cross-sectional study of retinal parameters in Huntington's disease and their evaluation as marker of disease progression.

### Clinical rationale for the study

Huntington's disease (HD) is a neurodegenerative disorder with dominant motor and neuropsychiatric symptoms. Involvement of sensory functions in HD has been investigated, however studies of retinal pathology are incongruent. Effect sizes of previous findings were not published. OCT data of the subjects in previous studies have not been published. Additional examination of structural and functional parameters of retina in larger sample of patients with HD is warranted.

### Materials and methods

This is a prospective cross-sectional study that included: peripapillary retinal nerve fiber layer thickness (RNFL) and total macular volume (TMV) measured by spectral domain optical coherence tomography (OCT) of retina, Pelli-Robson Contrast Sensitivity test, Farnsworth 15 Hue Color discrimination test, ophthalmology examination and Unified Huntington's disease Rating Scale (UHDRS). Ninety-four eyes of 41 HD patients examined in total 47 visits and 82 eyes of 41 healthy controls (HC) examined in total 41 visits were included. Analyses were performed by repeated measures linear mixed effects model with age and gender as covariates. False discovery rate was corrected by Benjamini-Hochberg procedure.

### Results

HD group included 21 males and 20 females (age 50.6±12.0 years [mean ± standard deviation], disease duration 7.1±3.6 years, CAG triplet repeats 44.1±2.4). UHDRS Total Motor

the GitHub repository (https://github.com/PavelDusek/hd-oct).

**Funding:** Czech Science Foundation project number and number 19–01747S Ministry of Health of the Czech Republic project number AZV–NU20–04–0136 Joint Programme – Neurodegenerative Disease Research (JPND) project number 8F19004 National Institute for Neurological Research (Programme EXCELES, ID Project No. LX22NPO5107) Funded by the European Union – Next Generation EU Charles University: Cooperatio Program in Neuroscience General University Hospital in Prague project MH CZ-DRO-VFN64165. The funders had no role in study design, data collection and analysis, decision to publish, or preparation of the manuscript.

**Competing interests:** The authors have declared that no competing interests exist.

Score (TMS) was 30.0±12.3 and Total Functional Capacity 8.2±3.2. Control group (HC) included 19 males and 22 females with age 48.2±10.3 years. There was no statistically significant difference between HD and HC in age. The effect of the disease was not significant in temporal segment RNFL thickness. It was significant in the mean RNFL thickness and TMV, however not passing false discovery rate adjustment and with small effect size. In the HD group, the effect of disease duration and TMS was not significant. The Contrast Sensitivity test in HD was within normal limits and the 15-hue-test in HD did not reveal any specific pathology.

## Conclusions

The results of our study support possible diffuse retinal changes in global RNFL layer and in macula in Huntington's disease, however, these changes are small and not suitable as a biomarker for disease progression. We found no other structural or functional changes in retina of Huntington's disease patients using RNFL layer and macular volume spectral domain OCT and Contrast Sensitivity Test and 15-hue-test.

## Clinical implications

Current retinal parameters are not appropriate for monitoring HD disease progression.

## 1. Introduction

Huntington's disease (HD) is an autosomal dominant hereditary neurodegenerative disorder caused by CAG repeat expansion in the huntingtin (*HTT*) gene encoding the HTT protein. HD manifests itself dominantly by motor, neuropsychiatric and cognitive symptoms. In the last decades, a substantial number of studies examining sensory functions in various neurodegenerative diseases including HD were published.

There is evidence of the presence of HTT in retina and the impairment of retinal functions in various animal models of HD. Helmlinger et al [1] found strong deficiencies in vision, retinal dystrophy, and mutant huntingtin in the retina of R6 transgenic mice. Johnson et al [2] showed reduced oscillatory potential amplitudes and disinhibition of the photopic response and accumulation of huntingtin in the horizontal cells of the retina of HD rats.

Several studies have indicated an impairment of visual evoked potentials (VEP) [3], retinal increment thresholds for a foveal blue test light [4], impaired color differentiation (CD) [5], an impairment of contrast sensitivity (CS) for moving gratings [6] and optical coherence tomography (OCT) parameters [7–9] in HD humans. However, other studies reported no significant findings in OCT parameters [10] or OCT angiography [11]. All these studies included small number of participants or were methodologically insufficient, see S1 Table for details. Additionally, patients with HD do not report any kind of visual difficulties. Therefore, we decided to study both structural and functional parameters of retina in larger sample of patients with HD and compare them with normal controls.

## 2. Materials and methods

We recruited 44 patients with HD from a cohort of patients followed at our university HD clinic, to participate in the study, four of the patients were measured twice during the study period. This study protocol was reviewed and approved by ethical committee of General

University Hospital in Prague IORG0002175 and all subjects gave written informed consent prior to the enrolment to the study.

Inclusion criteria were as follows: genetically confirmed HD patients (CAG triplet count 35 or more) with Unified Huntington's disease Rating Scale Total Motor Score greater than 5 and older than 18 years of age willing to participate in the study between 3rd of October 2012 and 18th of April 2018.

Exclusion criteria were as follows: history of central nervous system disorder other than HD, inflammatory disorder of the eye in the last 3 months, history of optic neuritis, glaucoma, diabetic retinopathy, age-related macular degeneration, retinal and macular oedema, macular holes, vitreomacular traction syndrome, retinoschisis, retinal detachment, retinal neovascularization, and tumors. Patients were assessed by Unified Huntington's disease Rating Scale (UHDRS). Ophthalmological examination included best corrected visual acuity (100% contrast), intraocular pressure measurement and fundus examination. Pelli-Robson Contrast Sensitivity test Chart 4K, which uses a single large letter size (20/60 optotype) with contrast varying across groups of letters, was used to evaluate CS. A Pelli-Robson score of 1.5 to 2.25 indicates normal CS, a score of 0.9 to 1.35 indicates moderate contrast loss, a score of 0.3 to 0.75 severe contrast loss and a score of 0.15 and less indicates profound contrast loss. Farnsworth D-15 Color test was performed to evaluate CD. The results of the Farnworth D-15 test determine color perception or defects in deutan, protan or tritan discrimination.

Using a spectral-domain OCT machine (Heidelberg Spectralis), without pupil dilation, we obtained a circular scan manually centered on the optic nerve head (diameter 3.4mm, Automated Real-Time—ART 100) and a macular volume scan centered on the fovea (63 lines, ART 20). The thickness of the peripapillary retinal nerve fiber layer (RNFL) in μm and total macular volume (TMV) in $mm^3$ was automatically determined by the instrument (Heyex version 5.8). All retinal images were checked for scan quality. We analyzed the mean peripapillary RNFL thickness (RNFL-G) as the main measure of axonal health in the retina and the thickness of RNFL in the temporal peripapillary segment (RNFL-T) that represents the most vulnerable retinal fibers. We also used total macular volume (TMV) measured in the area of a 6 mm wide circular mask to represent global retinal changes.

As a control group, we selected 41 healthy subjects matched by age and gender from the clinic database of OCT measurements.

Study data were collected and managed using REDCap electronic data capture tool [12]. The data cleaning procedure was done using SciPy (Python for scientific computing) version 1.0.0 [13] and statistical analysis was done in R version 4.0.3. Inter-group difference was assessed as a hierarchical model of a subject consisted of two eyes (and repeated measurements in time in 4 cases) using repeated measures linear mixed-effects model (library lmerTest version 3.1.3 [14]) with group, age, and gender as fixed effects and subject as a random effect. Effect size was calculated as Hedges' g (library effectsize version 0.6.0.1 [15]). Figures were generated using library ggplot2 version 3.4.0. Within HD group, we assessed the effect of UHDRS TMS and disease duration using repeated measures linear mixed-effects model with subject as a random effect. False discovery rate was corrected by Benjamini-Hochberg procedure, results were considered significant with adjusted P lower than 0.05.

## 3. Results

Forty-one patients from our cohort of 44 HD patients completed the study. Three patients were excluded from the study due to vitreoretinal pathology (because of possible distortion of OCT measurements). Measurements of HD patients were compared with 41 healthy controls (HC). Demographic data, UHDRS Total Motor Score (TMS) and Total Functional Capacity

**Table 1. Demographic data in the Huntington's disease group.**

| Demographic parameter | Value |
|---|---|
| Number of HD patients (male: female) | 41 (21:20) |
| Mean age in years (±SD) | 50.6 (±12.0) |
| Mean disease duration in years (±SD) | 7.1 (±3.6) |
| Mean CAG triplet repeats (±SD) | 44.1 (±2.4) |
| Mean UHDRS TMS (±SD) | 30.0±12.3 |
| Mean UHDRS TFC (±SD) | 8.2±3.2 |

SD–standard deviation, TFC–Total Functional Capacity, TMS–Total Motor Score, UHDRS–Unified Huntington's Disease Rating Scale

(TFC) listed in Table 1. The age and gender of HC group did not differ from the HD group (Mann-Whitney U-test p = 0.369 and Fisher's exact test p = 0.8253 respectively). Summary of the OCT measurements and effect sizes of the intergroup difference are listed in Table 2.

The linear mixed effects models for global mean RNFL thickness (RNFL-G) and for TMV showed significant difference between HD patients and HC (P = 0.0272, P = 0.03894 respectively), however, this effect did not pass the false discovery rate adjustment (P = 0.0576, P = 0.06814 respectively) and the effect size was small. The model for temporal segment RNFL thickness (RNFL-T) was not significant. Interestingly, there was a very significant effect of gender on the macular volume. All P-values of the three intergroup models are listed in Table 3. Effect sizes of intergroup differences for RNFL-T, RNFL-G and TMV are listed in Table 3. Mean TMV in HD patients was $8.5\pm0.37$ mm$^3$.

We analyzed the effect of disease duration and UHDRS TMS on RNFL thickness and TMV within the HD group also using linear mixed effects models. The effect of UHDRS TMS on temporal segment RNFL and on TMV was significant. None of these within group effects passed the false discovery rate adjustment. Other effects were not significant. All P-values of the three within HD group models are listed in the Table 4.

Pelli-Robson Contrast Sensitivity test Chart 4K was performed in 13 HD patients. Contrast sensitivity for both eyes was 1.64±0.04, which is within normal limits. Farnsworth D-15 Color test was performed in 14 HD patients. 6 patients had unspecified pathology, 6 had normal results, 1 patient had tritanopia and 1 patient had deuteranopia.

## 4. Discussion

There are a few studies proving the presence of mutant HTT in retina in animal HD models [1, 2]. Unfortunately, there is no post-mortem human study of retina in HD patients.

Even in long term clinical observation, HD patients report no subjective complaints about vision. On the other hand, several studies using various paraclinical tools (VEP, OCT etc.) found evidence for structural changes of retina or a functional impairment of visual pathways

**Table 2. RNFL-G thickness, RNFL-T thickness and macular volume parameters in Huntington's disease patients and in healthy controls.**

| OCT parameter | HD patients (mean ± SD) | Control group (mean ± SD) | Hedges' g (0.95-CI) |
|---|---|---|---|
| RNFL thickness G | 96.7±7.7 µm | 101±8.8 µm | 0.475 (0.157–0.792) |
| RNFL thickness T | 69.6±10.4 µm | 71.1±11.1 µm | 0.138 (-0.175–0.451) |
| Total Macular volume | 8.58±0.387 mm3 | 8.71±0.423 mm3 | 0.317 (-0.0145–0.647) |

RNFL–Peripapillary Retinal Nerve Fiber Layer, G–global segment, T–temporal segment, SD–standard deviation, CI–confidence interval.

**Table 3. Linear mixed effects models for RNFL-G thickness, RNFL-T thickness, and macular volume with group, age and gender as fixed effects.**

| Response OCT variable | Fixed effect | P-value | Adjusted P-value |
|---|---|---|---|
| RNFL-G thickness | Group (HD vs. HC) | **0.0272** | 0.0575 |
| RNFL-G thickness | Age | 0.220 | 0.281 |
| RNFL-G thickness | Gender | 0.700 | 0.735 |
| RNFL-T thickness | Group (HD vs. HC) | 0.403 | 0.470 |
| RNFL-T thickness | Age | **0.013** | **0.0346** |
| RNFL-T thickness | Gender | 0.762 | 0.762 |
| Total Macular Volume | Group (HD vs. HC) | **0.0389** | **0.068** |
| Total Macular Volume | Age | 0.0548 | 0.0886 |
| Total Macular Volume | Gender | **0.00154** | **0.004642** |

RNFL–Peripapillary Retinal Nerve Fiber Layer, OCT–optical coherence tomography, G–global segment, T–temporal segment, HD–Huntington's Disease, HC–healthy controls.

in HD patients [3–9]. However, a comparison of results between individual studies reveals conflicting findings in various parameters.

Total macular volume (TMV) was reduced in HD patients in study by Haider et al [8], but there was no difference between HD an HC in studies by Kersten et al [7], Andrade et al [10]. Our study showed significant difference (not passing the false discovery rate adjustment), but with small effect size, which may explain the discrepancy. To further confirm that the significant difference is indeed a false positive and false discovery rate adjustment was appropriate, we performed a post-hoc analysis. A correlation analysis of TMV and age (Pearson's r = -0.24, P = 0.04) or triplet expansion length (Pearson's r = -0.15, P = 0.20) in the patients group shows that changes of TMV are driven by aging only and not by changes associated with Huntington's disease. To evaluate effect of Huntington's disease on accelerated macular aging, we created a linear mixed-effects model with interaction of age and disease effects, however, this interaction was insignificant (P = 0.84). As can be noted in the Fig 1, the rate of macular loss with aging is even higher (not significantly) in the control group, which contradicts the notion that Huntington's disease could accelerate the macular aging.

Additionally, there was a very significant effect of gender on TMV. As previously described in the literature [16], changes in male maculae are on average greater than in female maculae, and we have reproduced these results.

Mean total RNFL thickness was reduced in HD patients with Total Functional Score Stage III in study by Gatto et al [9] but there was no difference in studies by Kersten et al [7], Haider et al [8], Andrade et al [10], and Di Maio [11]. Our study showed significant difference (again

**Table 4. Linear mixed effects models for RNFL-G thickness, RNFL-T thickness, and macular volume in the Huntington's disease group with UHDR TMS and disease duration as fixed effects.**

| Response OCT variable | Fixed effect | P-value | Adjusted P-value |
|---|---|---|---|
| HD RNFL-G thickness | UHDRS TMS | 0.0776 | 0.116 |
| HD RNFL-G thickness | Disease Duration | 0.524 | 0.580 |
| HD RNFL-T thickness | UHDRS TMS | **0.0294** | 0.0576 |
| HD RNFL-T thickness | Disease Duration | 0.152 | 0.213 |
| HD Total Macular Volume | UHDRS TMS | **0.0302** | 0.0576 |
| HD Total Macular Volume | Disease Duration | 0.228 | 0.281 |

RNFL–Peripapillary Retinal Nerve Fiber Layer, OCT–optical coherence tomography, G–global segment, T–temporal segment, HD–Huntington's Disease, UHDRS–Unified Huntington's Disease Rating Scale, TMS–Total Motor Score.

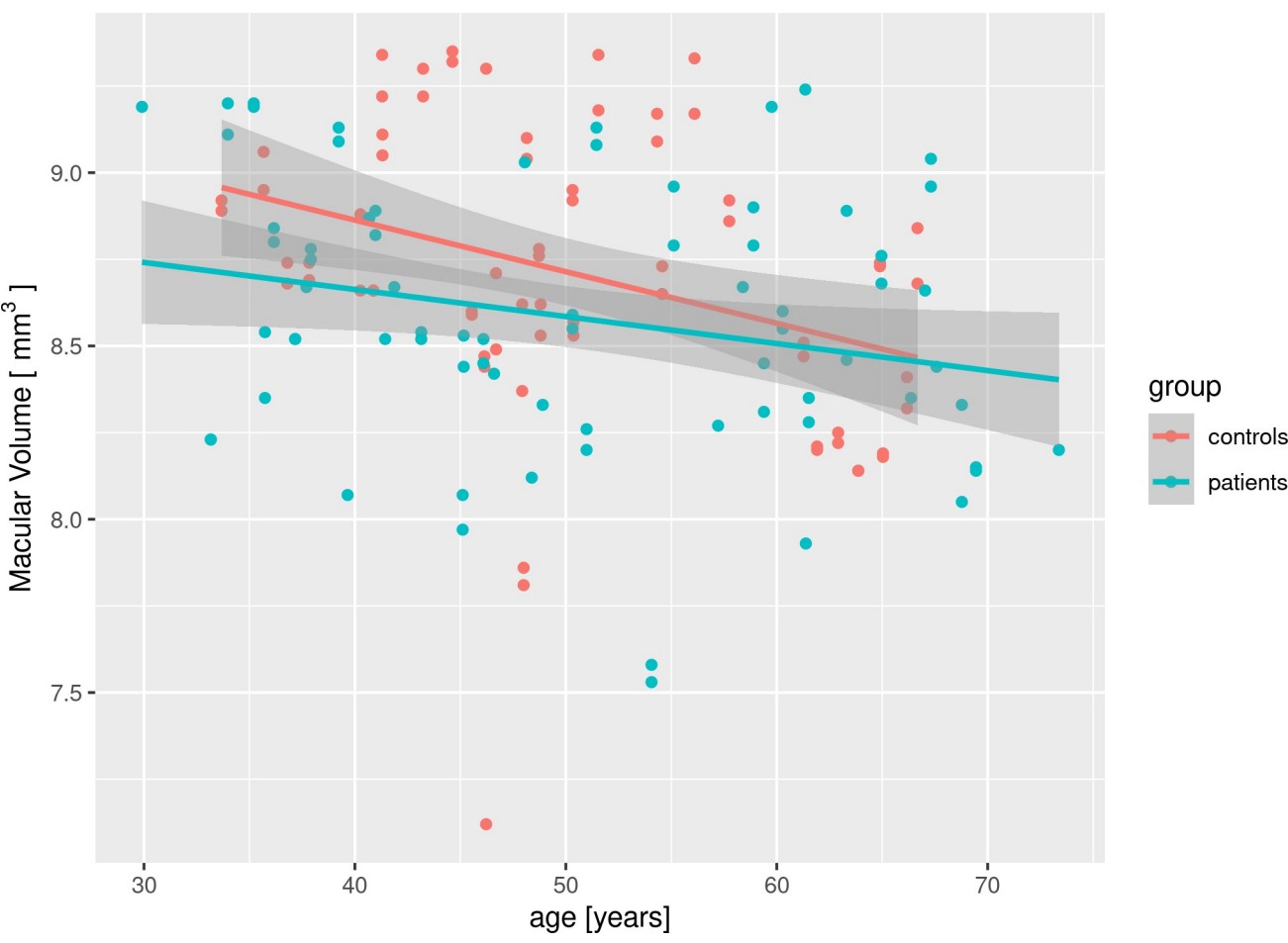

**Fig 1. Scatter plot of macular volume vs. age in patient and control group.** Note that the rate of macular volume loss with aging is steeper in the control group than in the patient group. The difference in slopes is not significant.

not passing the false discovery rate adjustment), but with small effect size, which may, again, explain the discrepancy.

Temporal RNFL thickness was hypothesized to be the part of RNFL most vulnerable to neurodegeneration [9]. Kersten et al [7] and Gatto et al [9] found it reduced in HD patients but there was no difference in the study by Haider et al [8]. Our study didn't show a significant intergroup difference. However, there was a significant effect of age in the intergroup model. Additionally, we found a significant effect of UHDRS TMS in the within HD group model, but this effect did not pass the false discovery rate adjustment. Our results support the notion that temporal RNFL atrophy is driven by age (rather than disease), which could explain the incongruity of the studies with different populations.

Peripapillary RNFL thickness was not different in HD patients from healthy controls in studies by Kersten et al [7] and Haider et al [8]. Inferior and nasal RNFL thickness were not altered by HD in the study by Gatto [9], were not examined in other studies and are not reported as significant to neurodegeneration. Therefore, we did not examine them to limit multiple statistical comparisons and increase statistical power of our study. Referenced studies, their sample size and their main findings are summarized and compared in S1 Table.

In our prospective cross-sectional study, we examined 82 eyes–the largest study sample on the topic. In 41 HD, we measured both retinal structural and visual functional parameters, six

subjects were examined twice leaving some possibility of longitudinal evolution of the parameters. We focused not only on the detection on the impairment but also on mutual relationship of functional and structural parameters. To our best knowledge, only single study by Kersten et al [7] investigated OCT and color discrimination together as in our study. Our findings in larger sample size did not confirm the results of the study by Kersten et al [7], who reported a negative correlation between macular volume and disease duration and UHDRS motor score. However, in Kersten study, there were no macular volume changes between HD and controls. It could be speculated that the macular volume changes may be effect of aging rather than effect of HD, since they are in both groups of subjects. We also cannot corroborate the correlation between temporal RNFL thickness and disease duration [7].

Furthermore, we found no pathology for contrast sensitivity evaluated by Pelli-Robson Contrast Sensitivity test Chart 4K in HD patients. This test examines the stationary CS, which reflects the functional integrity of parvocellular pathways, sensitive to high spatial frequencies and low temporal frequencies. O'Donnell et al [6] reported a selective deficit in CS for moving gratings in HD patients, dependent on the selective dysfunction in the magnocellular pathway, sensitive to low spatial frequencies, high temporal frequencies, and luminance.

Our results for color discrimination evaluated by Farnsworth D-15 Color test were nonspecific and ambiguous. Cognitive impairment may be an important factor interfering with the color discrimination. Büttner et al [5] found the mean total error scores and the partial scores for the red-green and the blue-yellow axes in Farnsworth-Munsell 100 Hue test which is more sensitive than our Farnsworth D-15 Color test. Kersten et al [7] found incorrect identification of Ishihara plates in HD patients, but they did not distinguish among types of color discrimination impairments. Ishihara plates method is used mainly for testing color discrimination in congenital disorders and may not be applicable in HD.

Haider et al [8] found no difference in ophthalmological findings in HD patients compared with HC, such as in our study.

## 5. Study limitations

Even though this study is the largest done on the topic, the sample size still may be underpowered to show significant difference. HD is a rare disease, and it is a challenge to reach larger group of patients willing to participate in the study. Additionally, if OCT and functional parameters correlated with disease progression, the findings would be most profound in later stages of the disease, when it is most difficult for the patients to undergo all possible examinations and endure long examination time. Therefore, we omitted some of the functional and structural examinations (VEP, foveal blue test, OCT angiography) that may have been stressful or exhausting for the study participants. Scotopic/photopic testing described in rats [2] was not performed. Only adult patients were included in the cohort and maximum triplet expansion length was 51, so these results are not transferable to juvenile Huntington's disease or late disease stages. Retinal ganglion cell complex was not analyzed. Analysis of the individual retinal layers of the macula requires a high-quality scan. Examination of the retina using OCT had to be adapted to involuntary movements and the ability to cooperate during examination in patients with HD. In many patients, due to the limitations of the current technology, the quality of the scans does not allow reliable segmentation of the retinal layer in the macula.

## 6. Conclusion

The results of our study support the notion that there are abnormalities in the temporal RNFL layer of Huntington's disease patients. However, it seems that, within the limits of the current technology and software, the magnitude of these abnormalities is very small and not clinically

useful. There were no other changes in the structural parameters of RNFL layer and total macular volume by means of spectral domain OCT and no changes in selected functional parameters in HD patients. According to our results, retinal spectral domain OCT at its present development for measurement of RNFL layer and total macular volume is not appropriate as a marker of HD progression. D-15 color testing is not a good marker of retinal changes associated with Huntington's disease. Surprisingly, our findings differ from most of the previous published studies. Prospective longitudinal studies and/or meta-analyses of all published studies are needed to solve this discrepancy. Additionally, further studies may focus more on other retinal layers, such as retinal ganglion cells.

## Supporting information

**S1 Table. Comparison of published studies and their findings in the main OCT parameters.** HD–Huntington's disease patients, HC–healthy controls
(DOCX)

## Author Contributions

**Conceptualization:** Jan Roth, Jana Lizrova Preiningerova.

**Data curation:** Pavel Dusek, Ales Kopal, Jana Lizrova Preiningerova.

**Formal analysis:** Pavel Dusek.

**Funding acquisition:** Jiri Klempir.

**Investigation:** Ales Kopal, Michaela Brichova, Olga Ulmanova, Jiri Klempir, Jana Lizrova Preiningerova.

**Methodology:** Jan Roth, Jana Lizrova Preiningerova.

**Project administration:** Jan Roth.

**Software:** Pavel Dusek.

**Supervision:** Jan Roth, Jana Lizrova Preiningerova.

**Writing – original draft:** Ales Kopal.

**Writing – review & editing:** Pavel Dusek.

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
