## [Decision Letter · Decision Letter 0]

19 Oct 2022

PONE-D-22-18702Is Retina Affected in Huntington’s Disease? Is Optical Coherence Tomography a good biomarker?PLOS ONE

Dear Dr. Dusek,

Thank you for submitting your manuscript to PLOS ONE. After careful consideration, we feel that it has merit but does not fully meet PLOS ONE’s publication criteria as it currently stands. Therefore, we invite you to submit a revised version of the manuscript that addresses the points raised during the review process.I also have to appologize for the long delay in coming back to you. We had severe delays in reviewers' responses and eventually only 1 reviewer completed their review.

Please find their comments at the end of this email.==============================

We look forward to receiving your revised manuscript.

Very best,

Andreas Neueder

Academic Editor

PLOS ONE

Journal Requirements:

Reviewers' comments:

Reviewer's Responses to Questions

**Comments to the Author**

1. Is the manuscript technically sound, and do the data support the conclusions?

Reviewer #1: Partly

2. Has the statistical analysis been performed appropriately and rigorously? 

Reviewer #1: Yes

3. Have the authors made all data underlying the findings in their manuscript fully available?

Reviewer #1: Yes

4. Is the manuscript presented in an intelligible fashion and written in standard English?

Reviewer #1: Yes

5. Review Comments to the Author

Reviewer #1: The authors evaluate retinal OCT findings as a possible biomarker for HD severity and progression. This is the largest number of patients included in a retinal evaluation of HD, and thus deserves a comprehensive analysis of the different layers, in addition to the RNFL and RGC complex. The temporal RNFL changes they report are small. And there is enough variability that 4 referenced studies were evenly divided in terms of these changes. The expansion repeats (~44) are in agreement with adult onset/complete penetrance, but do not take into consideration more severe/juvenile forms, nor does it consider late disease. However, this is a minor consideration, particularly since the authors are talking about identifying biomarkers for HD. To compare with expansion repeat numbers, age and disease duration, the authors use SD-OCT, contrast sensitivity (for acuity) and D15 color testing. Scotopic/photopic testing was not performed. This is of interest because the authors quote a previous functional study released only as an abstract (Johnson et al, 2012) which suggested inner retinal dysfunction that may be unassociated with RGCs.

Previous HD models have shown retinal degeneration, but it is unclear whether the changes in the early models (in the fly and mouse) are true indicators of the disease, since their expansion repeats were huge, and may have contributed to unusual retinal degeneration patterns similar to retinitis pigmentosa.

Previous reports have shown a decline in contrast sensitivity in HD patients, suggesting a loss of outer retinal, or other outer-retina-associated function, but the majority of studies have really focused on inner retinal function; in particular retinal ganglion cells (RGCs) and their axons The authors found a loss of RNFL thickness in the termporal region, but this was a trend, and not significant. They conclude that there are global RNFL changes, but not suitable as a biomarker, and that retinal parameters are not appropriate for monitoring HD disease progression. This is a considerable leap, since what they are really saying is that a) retinal SD-OCT at its present development is not suitable. B) the retinal parameters they looked at are not suitable. C) D-15 color testing is not a good damage marker of retinal parameter changes in HD. All are valuable considerations. I question then why they found changes in total macular volume in HD patients, which does seem to be significant, and which does suggest some associated retinal function besides RNFL. It also suggests that there may be changes in layers other than the RNFL.

In short, the authors need to utilize their data to perform a more detailed analysis of the different layers of the retina. This would make it more comprehensive, and more accurate for the conclusions that they draw (that OCT is useless in identifying HD retinal biomarkers).

6. PLOS authors have the option to publish the peer review history of their article (what does this mean?). If published, this will include your full peer review and any attached files.

Reviewer #1: No

---

## [Author Response · Author response to Decision Letter 0]

18 Dec 2022

Dear reviewers, 

we are pleased to read your comments and insights to our work. You have provided us with key feedback that help us do better research. We had to deal with some issues by stating them in the limitations section. However, we still think that our findings are worth as a publication, since it contradicts older studies and has the largest sample size. We have addressed the comments by making the conclusion more appropriate to our findings. 

Our response to comments is as follows: 

- "The authors evaluate retinal OCT findings as a possible biomarker for HD severity and progression. This is the largest number of patients included in a retinal evaluation of HD, and thus deserves a comprehensive analysis of the different layers, in addition to the RNFL and RGC complex. The temporal RNFL changes they report are small. And there is enough variability that 4 referenced studies were evenly divided in terms of these changes. The expansion repeats (~44) are in agreement with adult onset/complete penetrance, but do not take into consideration more severe/juvenile forms, nor does it consider late disease. However, this is a minor consideration, particularly since the authors are talking about identifying biomarkers for HD."

We added the fact that this study included only adult Huntington's disease patients to the study limitations. 

- "To compare with expansion repeat numbers, age and disease duration, the authors use SD-OCT, contrast sensitivity (for acuity) and D15 color testing. Scotopic/photopic testing was not performed. This is of interest because the authors quote a previous functional study released only as an abstract (Johnson et al, 2012) which suggested inner retinal dysfunction that may be unassociated with RGCs."

We added to the study limitations that the scotopic/photopic testing was not performed. 

- "Previous HD models have shown retinal degeneration, but it is unclear whether the changes in the early models (in the fly and mouse) are true indicators of the disease, since their expansion repeats were huge, and may have contributed to unusual retinal degeneration patterns similar to retinitis pigmentosa. 

Previous reports have shown a decline in contrast sensitivity in HD patients, suggesting a loss of outer retinal, or other outer-retina-associated function, but the majority of studies have really focused on inner retinal function; in particular retinal ganglion cells (RGCs) and their axons The authors found a loss of RNFL thickness in the termporal region, but this was a trend, and not significant. They conclude that there are global RNFL changes, but not suitable as a biomarker, and that retinal parameters are not appropriate for monitoring HD disease progression. This is a considerable leap, since what they are really saying is that a) retinal SD-OCT at its present development is not suitable. B) the retinal parameters they looked at are not suitable. C) D-15 color testing is not a good damage marker of retinal parameter changes in HD. All are valuable considerations."

We have changed the conclusion section accordingly. 

- "I question then why they found changes in total macular volume in HD patients, which does seem to be significant, and which does suggest some associated retinal function besides RNFL. It also suggests that there may be changes in layers other than the RNFL."

We performed a post-hoc analysis showing that the difference in macular volume was a false positive and was appropriately corrected by the false discovery rate correction. We showed that there is no interaction between age and triplet expansion length in the effect on macular volume. The rate of macular volume loss with aging is even (not significantly) steeper in the control group than in the patients group. These analyses were added to the discussion section. 

- "In short, the authors need to utilize their data to perform a more detailed analysis of the different layers of the retina. This would make it more comprehensive, and more accurate for the conclusions that they draw (that OCT is useless in identifying HD retinal biomarkers)."

Analysis of the individual retinal layers of the macula requires a high-quality scan. Examination of the retina using OCT had to be adapted to involuntary movements and the ability to cooperate during the examination in patients with HD. In many patients, the quality of scans does not allow reliable segmentation of the retinal layers in the macula – because of their involuntary movements.  

In any case, we think that the examination that we cannot reliably perform in these patients is not a suitable method for studying the progression of the disease. 

We have changed the limitations section to state these issues and the conclusion section accordingly to make statements about the performed methods only. 

All the changes in the text were highlighted in yellow. 

Additionally, during the review process, funding of our work changed, so we edited the funding sources slightly.  

Thank you for reviewing our work. We look forward to your opinion on our changes. 

Yours faithfully, 

Pavel Dusek

---

## [Decision Letter · Decision Letter 1]

1 Feb 2023

PONE-D-22-18702R1

Is Retina Affected in Huntington’s Disease? Is Optical Coherence Tomography a good biomarker?

PLOS ONE

Dear Dr. Dusek,

Thank you for submitting your manuscript to PLOS ONE. After careful consideration, we feel that it has merit but does not fully meet PLOS ONE’s publication criteria as it currently stands. Therefore, we invite you to submit a revised version of the manuscript that addresses the points raised during the review process.

Please revise your paper to state that it is difficult to assess Huntington disease patients using current OCT technology, not that OCT cannot be used to evaluate these patients. The expert reviewer has indicated where changes should be made in your paper in the attached "reviewer's comments"

We look forward to receiving your revised manuscript.

Kind regards,

Alfred S Lewin, Ph.D.

Section Editor

PLOS ONE

Journal Requirements:

Reviewers' comments:

Reviewer's Responses to Questions

**Comments to the Author**

1. If the authors have adequately addressed your comments raised in a previous round of review and you feel that this manuscript is now acceptable for publication, you may indicate that here to bypass the “Comments to the Author” section, enter your conflict of interest statement in the “Confidential to Editor” section, and submit your "Accept" recommendation.

Reviewer #1: All comments have been addressed

2. Is the manuscript technically sound, and do the data support the conclusions?

Reviewer #1: Yes

3. Has the statistical analysis been performed appropriately and rigorously? 

Reviewer #1: Yes

4. Have the authors made all data underlying the findings in their manuscript fully available?

Reviewer #1: Yes

5. Is the manuscript presented in an intelligible fashion and written in standard English?

Reviewer #1: Yes

6. Review Comments to the Author

Reviewer #1: The authors have replied to my review with some limited changes. My major concern is that it is important to recognize that while the SD-OCT test has wide capabilities, it is limited by the problems of the current ability to examine. I have also evaluated HD patients, and their shake/eye movements make it difficult to get a ‘lock on’ and high quality scans. Thus, it is necessary to distinguish the problems of CURRENT technology, and not simply say that OCT cannot be used (it cannot be used without more refinements). Otherwise, people may simply quote their paper to show ‘it cannot be done’, and suppress future research. Thus the suggested changes:

abstract: in conclusions: ‘Current’, not ‘TheseDiscussion/third paragraph: add ‘**Changes** in male maculae are….Discussion/last paragraph/last sentence: ‘In many patients**, due to the limitations of the current technology**, the quality of the scans….Conclusions:However, it seems that, **within the limits of current technology and software**,**********

7. PLOS authors have the option to publish the peer review history of their article (what does this mean?). If published, this will include your full peer review and any attached files.

Reviewer #1: **Yes: **Steven L. Bernstein

---

## [Author Response · Author response to Decision Letter 1]

7 Feb 2023

Dear prof. Bernstein,

we thank you for your time and for your suggested changes to improve our research. We incorporated them fully into the manuscript:

 1. In the Abstract, we changed ‘These retinal parameters are not appropriate for monitoring HD disease progression.’ to ‘Current retinal parameters are not appropriate for monitoring HD disease progression.’

 2. In the Discussion, we changed ‘Male maculae are on average greater than female maculae.’ to ‘Changes in male maculae are on average greater than in female maculae.’

 3. In the Study Limitations, we changed ‘In many patients, the quality of the scans does not allow reliable segmentation of the retinal layer in the macula.’ to ‘In many patients, due to the limitations of the current technology, the quality of the scans does not allow reliable segmentation of the retinal layer in the macula.’

 4. In the Conclusion, we changed ‘However, it seems that the magnitude of these abnormalities is very small and not clinically useful.’ to ‘However, it seems that, within the limits of the current technology and software, the magnitude of these abnormalities is very small and not clinically useful.’

Additionally, we included the Figure 1 description into the manuscript file, as requested by the Editorial Office.

We think that the revised manuscript is now more balanced in terms of study limitations and possible suggestions for future research. We look forward to your opinion on the manuscript.

Yours sincerely,

Pavel Dusek

---

## [Editor Report · Decision Letter 2]

9 Feb 2023

Is Retina Affected in Huntington’s Disease? Is Optical Coherence Tomography a good biomarker?

PONE-D-22-18702R2

Dear Dr. Dusek,

We’re pleased to inform you that your manuscript has been judged scientifically suitable for publication and will be formally accepted for publication once it meets all outstanding technical requirements.

Kind regards,

Alfred S Lewin, Ph.D.

Section Editor

PLOS ONE
---

## [Editor Report · Acceptance letter]

14 Feb 2023

PONE-D-22-18702R2 

Is Retina Affected in Huntington’s Disease? Is Optical Coherence Tomography a good biomarker? 

Dear Dr. Dusek:

I'm pleased to inform you that your manuscript has been deemed suitable for publication in PLOS ONE. Congratulations! Your manuscript is now with our production department. 

Kind regards, 

on behalf of

Dr. Alfred S Lewin 

Section Editor

PLOS ONE